# Segment, Associate, and Classify: Decoupled Audio-Visual Segmentation Framework

## Abstract

The audio-visual segmentation task aims to segment sounding objects associated with the corresponding audio in visual data. Unlike previous supervised approaches, this paper presents a method that does not require ground-truth audio-visual masks during training. The proposed framework consists of three decoupled stages: (1) **segmenting** category and audio-agnostic objects solely from an input image, (2) **associating** input audio and segmented object masks to obtain the corresponding mask to the audio, and (3) **classifying** the object mask. We leverage the pretrained segmentation and vision-language foundation models in the segmentation and classification stages, respectively, and the audio-mask association module in the second stage is trained without relying on ground-truth correspondence between audio and object masks via a multiple-instance contrastive learning scheme. In the association module, we propose object mask representation to incorporate the local and global information of the objects and training framework to enhance the segmentation performance on the multi-source audio inputs. Our approach significantly outperforms previous unsupervised and weakly-supervised sound source localization and segmentation methods. Furthermore, our approach achieves a comparable performance to the supervised audio-visual semantic segmentation baseline.

## 1 Introduction

Associating the sounding object in visual data with a corresponding audio signal is one of the fundamental tasks in the multimodal understanding field. This task requires fine-grained alignment between data captured from different sensors. Humans possess the ability to perform this association between input audio and visual data through the tight association between observed visual and auditory signals in the natural world without explicit ground-truth correspondence.

This goal, which involves segmenting sounding objects from corresponding audio and visual data, has been approached from two different perspectives: sound source localization and audio-visual segmentation. The former task is addressed by approaches (Chen et al., 2021; Mo & Morgado, 2022a;b;b; Park et al., 2023; Sun et al., 2023; Senocak et al., 2023; Park et al., 2024) which leverage unlabeled audio-image datasets (Chen et al., 2020; Senocak et al., 2018) and train models in an unsupervised manner. However, since the audio is associated with coarse grid-level visual features, these approaches can only localize the rough position of sounding objects in the image. On the other hand, the approaches (Zhou et al., 2022; Mao et al., 2023; Liu et al., 2023b; Huang et al., 2023; Mo & Tian, 2023; Wang et al., 2024a; Liu et al., 2024a) to tackle the latter segmentation task utilize ground-truth sounding object masks during training to estimate fine-grained pixel-level audio-visual masks. Recently, the audio-visual segmentation task has been extended to the audio-visual *semantic* segmentation task (Zhou et al., 2023), which aims to estimate audio-visual semantic masks that provide pixel-level category information of sounding objects. However, annotating sounding object masks (with category information) in a video is extremely time-consuming, as annotators must listen to the audio while watching the corresponding video and draw object masks frame-by-frame. Therefore, scaling up the training dataset with the annotation is infeasible.

We aim to harness the benefits of both perspectives. Specifically, we train the model without ground-truth audio-visual masks while segmenting the sounding objects at a pixel level. This presents a challenge, as the model must learn the fine-grained association between the audio signal and the

pixel-level visual information without explicit ground-truth supervision. To address this challenge, we capitalize on the recent significant advancements in vision foundation models (Radford et al., 2021; Singh et al., 2022; Kirillov et al., 2023; Ke et al., 2023). Since these models have been trained on extremely large-scale datasets, endowing them with exceptional zero-shot estimation capabilities across various data domains, it is imperative to fully harness the capabilities of pretrained vision foundation models for the audio-visual segmentation task.

In this paper, we introduce audio-visual segmentation and semantic segmentation framework, *Segment, Associate, and Classify* (SeAC), which decouples the tasks into three distinct stages: (1) segmenting audio and category-agnostic object masks solely from an input image, (2) associating a set of object masks with input audio to establish correspondence, and (3) classifying the object masks by assigning the object category to the detected object masks (only necessary for semantic segmentation). Specifically, in the first stage, we detect and segment objects in the image using the segmentation foundation models (Kirillov et al., 2023; Ke et al., 2023). Since the object masks are detected solely from the images at the first stage, the masks include both sounding and non-sounding objects within the image. Therefore, in the second stage, the similarity between the input audio and a set of object masks is estimated by associating the audio and masks, and the audio-visual mask is obtained from the similarities. Finally, in the last stage, we assign category labels to each mask using vision-language models (Radford et al., 2021; Singh et al., 2022) to derive the audio-visual semantic mask.

In the framework, we present an unsupervised audio-mask association module to predict the similarity between input audio and detected object masks. The network is trained solely on pairs of audio and object masks, without any manual annotation of audio-visual masks. Since establishing correspondences between audio and sounding object masks during training is challenging due to the absence of ground-truth annotations, we employ a multiple-instance contrastive learning scheme, assuming that one of the detected object masks aligns with the corresponding audio signal. We propose a *local-global mask embedding representation* that incorporates the local and global visual features of the object masks. Moreover, we propose *multi-source audio-aware training*, which synthetically mixes multiple audios and maximizes the similarity between the mixed audio embedding and multiple mask embeddings in a contrastive loss.

Our contributions are summarized as follows:

- We propose the audio-visual segmentation and semantic segmentation framework, *Segment, Associate, and Classify* (SeAC), which decouples the tasks into three distinct stages: audio and category agnostic object segmentation, the association between audio and object masks, and mask classification. Through this decoupled framework and training the association module in an unsupervised manner, the framework can leverage the benefits from pretrained vision foundation models to segment the sounding objects at a pixel level with no ground-truth audio-visual masks during training.

- We propose to train the audio-mask association module in an unsupervised manner via a multiple-instance learning framework without ground-truth audio-visual masks. In the module, we propose local-global mask embedding representation and a multi-source audio-aware training scheme.

- The approach outperforms the prior state-of-the-art (SoTA) unsupervised and weakly-supervised sound source localization and segmentation approaches with a large margin (+12 and +19 points F1-score improvements on single-source and multi-source settings, respectively). Moreover, our method reaches the performance of a supervised semantic segmentation baseline.

## 2 PROPOSED FRAMEWORK (SEAC)

In this section, we introduce SeAC, our decoupled audio-visual segmentation framework. The following sections explain the overall framework (Section 2.1), the audio-mask association module (Section 2.2), the mask classification module for the audio-visual semantic segmentation task (Section 2.3), and the unsupervised training of the framework (Section 2.4).

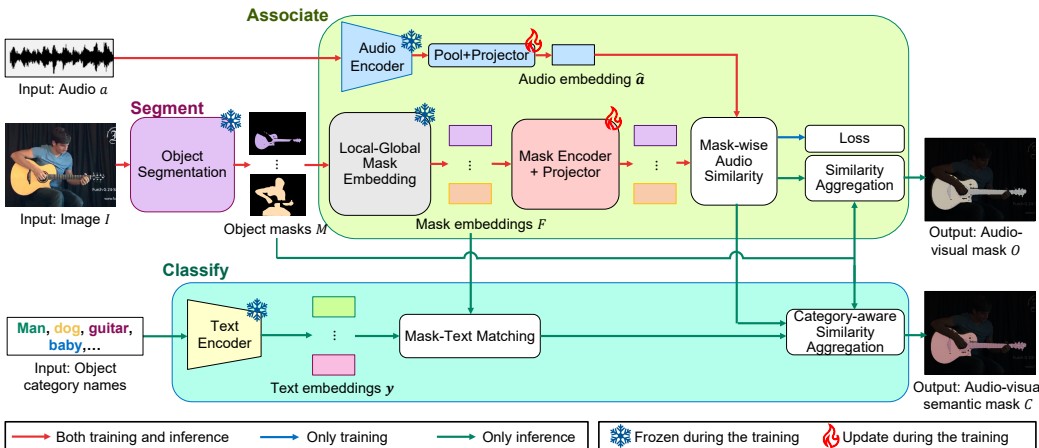

Figure 1: Overview of the proposed SeAC framework. In the *segmentation* stage, object masks are extracted solely from the input image. For the *association* stage, the detected object masks are associated with audio, and the audio-visual mask is estimated based on similarities between audio and the masks. During *classification*, the object category is assigned to each mask through matching between the text embeddings of the object category names and mask embeddings. The audio-visual semantic mask is generated using the audio-mask similarities and the assigned categories to object masks.

## 2.1 FRAMEWORK OVERVIEW

The pipeline of our framework is depicted in Fig. 1. The tasks are to estimate the audio-visual mask $O$ and audio-visual semantic mask $C$ from input image $I$ and audio $a$. The framework consists of three distinct stages: (1) object segmentation, (2) association between audio and object masks, and (3) mask classification. First, object masks are detected from the input image. Since this segmentation is not conditioned on the audio input, these masks include the sounding objects and the sound-irrelevant or background objects. In the audio-mask association stage, the input audio and set of object masks are encoded into embeddings, and the similarities between the audio embedding and the set of object mask embeddings are calculated. The similarities are used to train the networks in the audio-mask association module. During the inference, the similarities between audio and object masks are aggregated to predict the audio-visual mask. For the audio-visual semantic segmentation task, the category label is assigned for each mask using the text labels at the mask classification stage, and the audio-visual semantic mask is estimated.

**Object Segmentation.** The object segmentation stage consists of two steps: (1) detecting objects (top-$N$ confidence bounding boxes) in the image $I \in \mathbb{R}^{H \times W \times 3}$ using pretrained category-agnostic object detector (Maaz et al., 2022), and (2) using detected bounding boxes as an input prompt to the pretrained SAM (Kirillov et al., 2023) to obtain the $N$ binary object masks $M_n \in \{0, 1\}^{H \times W} (n = \{1, \dots, N\})$. Note that the audio signal is decoupled from the object segmentation task to leverage the pretrained segmentation foundation models.

## 2.2 AUDIO-MASK ASSOCIATION

This stage takes an input of audio and object masks detected from an image and predicts the similarity between the audio and the masks. The similarity is employed in the training (Section 2.4) and to estimate the audio-visual mask via *Similarity Aggregation* during the inference, explained later.

First, we extract embeddings from audio and a set of object masks. For the audio embedding extraction, following the previous works (Zhou et al., 2022; 2023), we convert audio waveform $a$ into a spectrogram via the short-time Fourier Transform (Griffin & Lim, 1984). The audio feature vector $a \in \mathbb{R}^{d_a}$ is extracted using pretrained VGGish network (Hershey et al., 2017), and attention pooling (Chen et al., 2021) along time-dimension. A set of object mask embeddings $F = \{f_0, \dots, f_N\} \in \mathbb{R}^{N \times d_m}$ is extracted from $N$ detected object binary masks and input RGB image $I$ via proposed global-local mask embedding representation, explained in below. These mask

embeddings are inputted into a mask encoder, which consists of a fully connected layer and Transformer (Vaswani et al., 2017) to consider the relation among mask embeddings in the image.

**Mask-wise Audio Similarity.** The audio and mask embeddings are then mapped into $d$-dimensional shared latent space using independent projection layers $\phi_a \in \mathbb{R}^{d_a \times d}$ and $\phi_m \in \mathbb{R}^{d_m \times d}$: $\hat{\boldsymbol{a}} = \phi_a(\boldsymbol{a})$ and $\hat{\boldsymbol{f}}_n = \phi_m(\boldsymbol{f}_n)$. The similarity $s_n$ between the audio embedding and $n$-th mask embedding is calculated as $s_n = \texttt{sim}(\hat{\boldsymbol{f}}_n, \hat{\boldsymbol{a}})$, where $\texttt{sim}(\cdot, \cdot)$ denotes the operation to calculate the cosine similarity between two vectors.

**Global-local Mask Embedding Representation.** To identify the sounding object from a set of object masks, it is essential to encode not only local (*e.g.*, each object's semantics) but also global (*e.g.*, the background contexts) information into mask embeddings. Therefore, we propose to represent mask embeddings that incorporate local and global visual information of the masks inspired by zero-shot referring image segmentation models (Yu et al., 2023; Bracha et al., 2023).

For each detected binary object mask $M_n$, we apply two transform operations to the input image $I$ using the object mask. Then, we encode two transformed images $I_n^L$ and $I_n^G$ per mask using vision encoder of the CLIP (Radford et al., 2021) $\psi^v$ and obtain the mask embedding $\boldsymbol{f}_n \in \mathbb{R}^{d_m}$ for $n$-th object mask as follows:

$$\boldsymbol{f}_n = \lambda \psi^v(I_n^L) + (1 - \lambda)\psi^v(I_n^G), \tag{1}$$

where $\lambda$ denotes the hyperparameter which balances the local and global embeddings. The transformation to obtain the local image representation $I_n^L$ is expressed as follows:

$$I_n^L = \mathcal{T}(M_n \odot I), \tag{2}$$

where $\odot$ is a Hadamard product operation, and $\mathcal{T}$ denotes the cropping operation using the bounding box of the mask. The transformation to obtain the global image representation $I_n^G$ with a Gaussian blur operation $\mathcal{B}$ is expressed as follows:

$$I_n^G = (1 - M_n) \odot \mathcal{B}(I) + M_n \odot I. \tag{3}$$

**Similarity Aggregation.** During the inference, we aggregate the set of mask-wise audio similarities by selecting the maximum similarity value per pixel location $(u, v)$ among object masks that are detected at that pixel. The output audio-visual mask $O \in \mathbb{R}^{H \times W}$, computed as follows:

$$O(u, v) = \max_{i=\{0,\ldots,N\}}(\hat{s}_i \cdot M_i(u, v)), \tag{4}$$

where $\hat{s}_i$ denotes the $[0, 1]$ ranged normalized mask-wise similarity among $N$ audio-mask similarities, *i.e.*, , $\hat{s}_i = (s_i - \min_i s_i)/(\max_i s_i - \min_i s_i)$.

## 2.3 MASK CLASSIFICATION

Following the zero-shot image understanding task setups (Radford et al., 2021; Li* et al., 2022; Li et al., 2022), we input the text prompts of sounding object category names in the target dataset with a template sentence to the CLIP's text encoder to obtain the $Y$ text embeddings $\mathcal{Y} = \{\boldsymbol{y}_1, \ldots, \boldsymbol{y}_Y\} \in \mathbb{R}^{Y \times d_m}$. The category of the $n$-th mask is assigned based on the maximum cosine similarity between the text embeddings and the mask embedding as follows: $t_n = \text{argmax}_{\boldsymbol{y}_i \in \mathcal{Y}}(\texttt{sim}(\boldsymbol{y}_i, \boldsymbol{f}_n))$. We extend the similarity aggregation to the audio-visual semantic segmentation task, *Category-aware Similarity Aggregation*, to estimate the audio-visual semantic mask.

**Category-Aware Similarity Aggregation.** The audio-visual semantic mask $C \in \mathbb{R}^{H \times W}$, in which each pixel contains the category index of the object if that object is sounding and 0 (background label) for non-sounding pixels, is estimated from the set of category labels $\boldsymbol{t} = \{t_0, \ldots, t_N\}$, the normalized similarities $\hat{\boldsymbol{s}}$, and set of masks $M$ as follows:

$$C(u, v) = \begin{cases} 1 + \boldsymbol{t}[\arg\max_i(\hat{s}_i \cdot M_i(u, v))] & \text{if } \max_i(\hat{s}_i \cdot M_i(u, v)) > \sigma, \\ 0 & \text{otherwise,} \end{cases} \tag{5}$$

where $\sigma$ is the threshold hyperparameter, and $\boldsymbol{t}[i]$ denotes an operation to extract $i$-th element in vector $\boldsymbol{t}$. This per-pixel operation assigns the object category label of the mask with the maximum similarity score if the maximum score is higher than the threshold and assigns the background label if the score is lower.

### 2.4 Unsupervised Multi-source Audio-aware Framework Training

We only train the modules in the association stage, such as the attention pooling layer, mask encoder, and two projection layers, and the weights of the pretrained models (*e.g.*, SAM, CLIP's text/image encoders, and VGGish) are fixed during the training. Since the existing large-scale audio-visual datasets (Gemmeke et al., 2017; Chen et al., 2020) mainly consist of audio and images with only a single sounding object[1], the training on these datasets may fail to infer when multiple objects are sounding in the image. Therefore, we propose to train the module in a multi-source audio-aware manner. Specifically, we mix multiple audio waveforms during the training to generate multi-source audio synthetically, and Multi-Source Audio-aware Multiple-Instance Contrastive Learning (MSA-MICL) loss maximizes the alignment between mixed audio and multiple mask embeddings in contrastive learning.

**Audio Mixing Augmentation.** We randomly divide a mini-batch into $K$ groups ($\mathcal{K} = \{\mathcal{K}_1, \ldots, \mathcal{K}_K\}$), and the audio waveforms within a group are synthetically mixed to generate an audio mixture waveform $a_k$ of $k$-th group as follows: $a_k = \Sigma_{i \in \mathcal{K}_k} a_i$. The embedding of the mixed audio $\hat{a}_k$ is also extracted in the same manner explained in Section 2.2.

**MSA-MICL Loss.** First, we briefly review the Multiple-Instance Contrastive Learning (MICL) loss, which is the similar loss design employed in EZ-VSL (Mo & Morgado, 2022a). In contrast to EZ-VSL, which applies MICL loss between the audio and grid-level image embeddings, we apply MICL loss between the audio and the set of object mask embeddings. MICL loss aligns the embeddings between the audio and the paired set of masks under the assumption that at least one of the object masks within a set of masks matches the corresponding paired (positive) audio while not matching the non-paired (negative) audio, *e.g.*, the audio from another sample in a mini-batch. More specifically, the alignment between the audio embedding and the most similar mask embedding in a positive set of masks is maximized, while the alignment between the audio embedding and the most similar mask embedding in a negative set of masks is minimized through as follows:

$$\mathcal{L}_{MICL}(S) = -\sum_{i=1}^{B} \log \frac{\exp(S_{i,i}/\tau)}{\Sigma_j^B \exp(S_{i,j}/\tau)} - \sum_{i=1}^{B} \log \frac{\exp(S_{i,i}/\tau)}{\Sigma_j^B \exp(S_{j,i}/\tau)}, \tag{6}$$

$$S_{i,j} = \max_{\hat{f}_n \in \hat{F}_i} (\text{sim}(\hat{f}_n, \hat{a}_j)), \tag{7}$$

where $B$ denotes the batch size, $S \in \mathbb{R}^{B \times B}$ is the cosine similarity matrix within a mini-batch, each element in $S$ ($S_{i,j}$) is the maximum cosine similarity between $j$-th audio embedding $\hat{a}_j$ and $i$-th set of mask embeddings $\hat{F}_i \in \mathbb{R}^{N \times d}$, and $\tau$ is a learnable temperature parameter.

If audios of the $i$-th and $j$-th samples in a mini-batch are mixed, the cosine similarities $S_{i,k}$ and $S_{j,k}$ calculated from the mixed audio embedding $\hat{a}_k$ and instances in each set of mask embeddings $\hat{F}_i$ and $\hat{F}_j$ should be maximized. Therefore, MICL loss is modified to consider the multiple positive samples, namely MSA-MICL loss, since the naive MICL loss minimizes the alignment between all the non-paired audio and the set of mask embeddings. The MSA-MICL loss maximizes the similarities between the mixed audio and multiple sets of mask embeddings, each of which corresponds to the original audio mixtures, as follows:

$$\mathcal{L}_{MSA-MICL}(S') = -\sum_{k=1}^{K} \log \frac{\Sigma_{i \in \mathcal{K}_k} \exp(S'_{i,k}/\tau)}{\Sigma_j^B \exp(S'_{j,k}/\tau)} - \sum_{i=1}^{B} \log \frac{\exp(S'_{i,\Omega(i)}/\tau)}{\Sigma_k^K \exp(S'_{i,k}/\tau)}, \tag{8}$$

where $S' \in \mathbb{R}^{B \times K}$ is the cosine similarity matrix between $K$ mixed audio embeddings and $B$ set of mask embeddings, and $\Omega(i)$ denotes the operation to obtain group index that sample $i$ belongs. Note that Eq. (8) is equivalent to Eq. (6) when $K = B$ (no augmentation applied).

## 3 Experiments

### 3.1 Experimental Setup

**Datasets.** Our framework is trained on the VGGSound (Chen et al., 2020) dataset, one of the large-scale datasets with corresponding audio and video pairs. Following previous audio-visual

---

[1]About 90% of audio data in the VGGSound dataset (Chen et al., 2020) are a single-source sound.

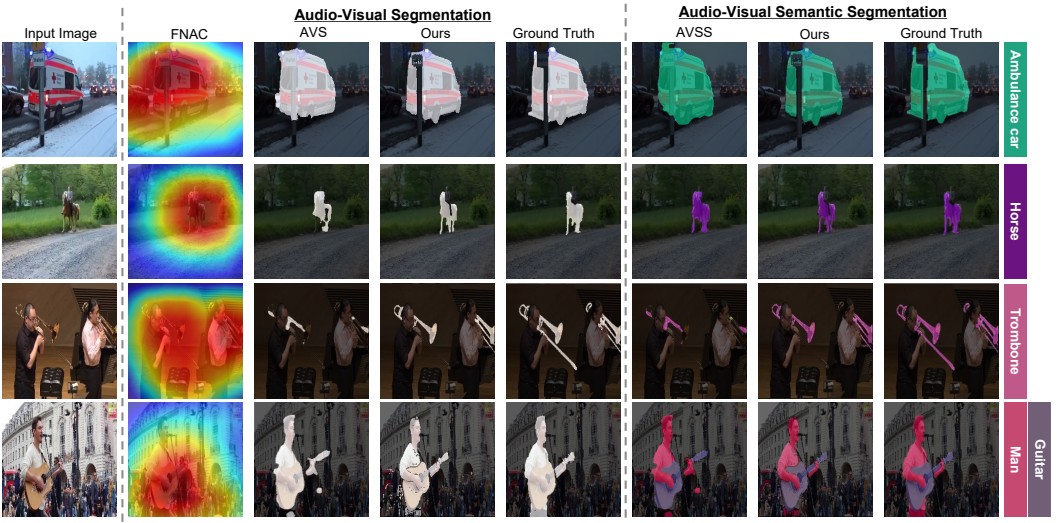

Figure 2: Example of detected object masks on AVSBench (Zhou et al., 2022). The detected object masks are visualized in different colors, and only 10 masks with high detection confidence values are shown for the visualization.

Figure 3: Qualitative results of the audio-visual segmentation and semantic segmentation.

localization approaches (Sun et al., 2023; Senocak et al., 2023; Park et al., 2024), we use a subset of 144k pairs of audio and videos. Note that we train the model only using paired audio and videos.

For the evaluation, we employ four commonly used datasets, such as AVSBench (Zhou et al., 2022), AVSBenchSemantics (Zhou et al., 2023), VGG-SS (Chen et al., 2021), and Extended VGG-SS (Mo & Morgado, 2022b). AVSBench includes binary segmentation masks indicating audio-visually related pixels and has two subsets: Single-source and Multi-source. The Single-source subset consists of videos in which a single-sounding object exists. On the other hand, the Multi-source subset consists of videos in which multiple-sounding objects exist, such as a baby crying while a dog is barking. AVSBenchSemantics includes ground-truth-sounding object masks with 70 object category annotations. We use test subsets in these benchmarks for the evaluation (740, 64, and 1554 videos in three sets, respectively). The VGG-SS evaluation dataset contains bounding box annotations of sound sources for around 5k samples, and the Extended VGG-SS dataset is used to verify the robustness against more edge cases, such as the cases when none of the objects are sounding or the sounding objects are not visible in the image.

**Evaluation Metrics.** Following the prior works (Zhou et al., 2022; Mo & Morgado, 2022b), we employ mean Intersection over Union (mIoU) and F1-score (F-score) for AVSBench and AVSBenchSemantics, Consensus Intersection over Union (cIoU) and Area Under Curve (AUC) for VGG-SS, and Average Precision (AP) and Max-F1 score for Extended VGG-SS dataset, respectively.

### 3.2 IMPLEMENTATION DETAILS

**Input Data Preprocessing.** We clip center 5-second videos from the original videos, and frames are extracted with 1 FPS to obtain audio and image pairs during the training. Since the videos in AVSBenchSemantics are 10-second videos, these videos are divided into two 5-second videos and preprocessed them. The audio waveform is resampled to 16kHz mono audio.

Table 1: Audio-visual segmentation comparison on Single-Source and Multi-Source subsets on AVSBench and VGG-SS/Extended VGG-SS benchmarks. ✗* denotes the weakly-supervised approaches that require the audio category label during the training.

| Method | Mask annotation | Single-Source mIoU/F-score | Multi-Source mIoU/F-score | VGG-SS cIoU/AUC | EXTVGG-SS AP/Max-F1 |
|---|---|---|---|---|---|
| AV-SAM (Mo & Tian, 2023) | | 40.47/56.57 | – | – | – |
| AVS (Zhou et al., 2022) | ✓ | 72.79/84.80 | 47.88/57.80 | 36.86/37.00 | |
| GAVS (Wang et al., 2024a) | | 80.06/90.20 | 63.70/77.30 | 41.07/41.10 | — |
| CAM (Zhou et al., 2016) | | 19.26/27.88 | 12.65/19.83 | – | – |
| C²AM (Xie et al., 2022) | | 30.87/36.55 | 25.33/29.58 | – | – |
| WS-AVS (Mo & Raj, 2023) | ✗* | 34.13/51.76 | 30.85/46.87 | – | – |
| MSSL (Qian et al., 2020) | | 44.89/66.30 | 26.10/36.30 | – | – |
| M2VSL (Mo & Wang, 2024) | | 37.85/55.21 | 35.26/49.35 | 46.80/50.20 | – |
| EZ-VSL (Mo & Morgado, 2022a) | | 26.43/29.20 | 21.36/22.50 | 35.96/38.20 | 24.55/30.90 |
| SLAVC (Mo & Morgado, 2022b) | | 28.10/34.60 | 24.37/25.56 | 37.79/39.40 | 32.95/40.00 |
| MarginNCE (Park et al., 2023) | | 33.27/45.33 | 27.31/31.56 | 38.25/39.06 | 30.58/36.80 |
| FNAC (Sun et al., 2023) | ✗ | 27.15/31.40 | 21.98/22.50 | 39.50/39.66 | 23.48/33.70 |
| Alignment (Senocak et al., 2023) | | 29.60/35.90 | – | 39.94/40.02 | 34.73/40.70 |
| ACL-SSL (Park et al., 2024) | | 59.76/69.03 | 41.08/46.67 | **49.46**/46.32 | **40.79**/49.10 |
| SeAC(Ours) | | **65.31/81.52** | **47.39/65.47** | 48.58/**48.68** | 40.54/**49.96** |

**Object Segmentation.** We input an image to a class-agnostic object detector, MViT (Maaz et al., 2022), with the text prompt "all objects" to detect objects in the input image. We use the pretrained weights provided in the official repository[2] which is trained on multiple object detection datasets (Lin et al., 2014; Krishna et al., 2017; Plummer et al., 2015). To remove the overlapping bounding boxes, we apply Non-Maximum Suppression with IoU=0.5. Then, we use top-$N(= 50)$ confidence bounding box coordinates as a prompt to SAM (Kirillov et al., 2023) with ViT-H backbone (Dosovitskiy et al., 2021) to generate the object masks conditioned on the bounding boxes. The detected masks are visualized in Fig. 2. The figure shows that multiple objects are detected and segmented. We empirically found that directly using SAM prompted with points uniformly distributed in the image failed to segment small objects or partially segment the object.

**Audio-Mask Association Module.** To obtain mask embeddings, we employ ResNet-50 as a CLIP visual encoder $\psi^v$ ($d_M = 1024$). The pretrained weights of the CLIP are obtained from the official repository[3] and are fixed during the training. We use the VGGish model (Hershey et al., 2017) pretrained on AudioSet (Gemmeke et al., 2017) for the audio encoder. The mixing weight $\lambda$ in Eq. (1) is set to 0.6, and set threshold parameter $\sigma$ in Eq. (5) to 0.9, empirically. Inspired by the curriculum learning framework (Bengio et al., 2009), we linearly increase the probability of applying audio mixing augmentation (Section 2.4) from 0.0 to 0.5 according to the epochs, and we fix the number of samples to be mixed to 2 ($K = B/2$) if the augmentation is applied. See the Appendix for further technical details and hyperparameters.

### 3.3 QUALITATIVE RESULTS

The qualitative results and the comparison with the audio-source localization approach (FNAC (Sun et al., 2023)) and the supervised audio-visual segmentation (AVS (Zhou et al., 2022)) and semantic segmentation (AVSS (Zhou et al., 2023)) approaches are summarized in Fig. 3. From the figure, our model segments the sounding objects from images at a pixel level without audio-visual mask annotations during the training, while the audio-source localization approach can only localize the sounding objects in the image. Moreover, our approach correctly assigns the category labels for the sounding objects for the audio-visual semantic segmentation task, even when multiple sounding objects exist in the image (last row).

### 3.4 COMPARISON TO PRIOR WORK

**Audio-Visual Segmentation.** Table 1 summarizes the quantitative results and comparison of the audio-visual segmentation on two subsets (Single-Source/Multi-Source) in the AVSBench, VGG-

---

[2] https://github.com/mmaaz60/mvits_for_class_agnostic_od
[3] https://github.com/openai/CLIP

Table 2: Audio-visual semantic segmentation comparison on AVSBenchSemantics.

| Method | Visual Backbone | Audio-visual semantic mask annotation | AVSS | |
|---|---|---|---|---|
| | | | mIoU (↑) | F-score (↑) |
| 3DC (Mahadevan et al., 2020) | ResNet-18 | | 17.27 | 21.60 |
| AOT (Yang et al., 2021) | ResNet-50 | ✓ | 25.40 | 31.00 |
| AVSS (Zhou et al., 2023) | ResNet-50 | | 20.18 | 25.20 |
| SeAC(Ours) | ResNet-50 | ✗ | 20.60 | 23.56 |
| | ViT-B/16 | | 25.52 | 29.59 |

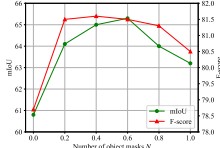
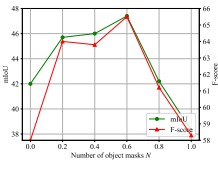
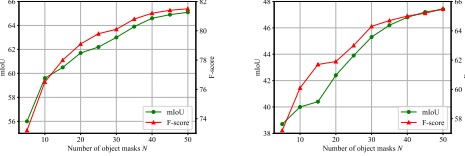

Figure 4: The ablation study of the mixing weight $\lambda$ in Eq. (1) (Left: Single-Source, Right: Multi-Source).

Figure 5: The ablation study of the number of input masks $N$ (Left: Single-Source, Right: Multi-Source).

SS, and Extended VGG-SS datasets. The comparison approaches we employ include weakly-supervised audio-agnostic object localization (Zhou et al., 2016; Xie et al., 2022), weakly-supervised audio-visual segmentation (Mo & Raj, 2023; Qian et al., 2020), unsupervised sound source localization (Chen et al., 2021; Mo & Morgado, 2022a;b;b; Park et al., 2023; Sun et al., 2023; Senocak et al., 2023; Park et al., 2024), and supervised audio-visual segmentation approaches (Zhou et al., 2022; Mo & Tian, 2023; Wang et al., 2024a). The supervised and weakly-supervised approaches are trained on the AVSBench (Zhou et al., 2022) since these approaches necessitate ground-truth audio-visual masks or category labels during training, respectively. Conversely, the unsupervised sound source localization approaches are trained on the same dataset (144k samples in VGGSound) as ours.

From the Table 1, the proposed method outperforms the weakly-supervised and unsupervised approaches by a substantial margin, especially on the F-score that measures the contour similarity. The F-score is improved from prior SoTA (ACL-SSL (Park et al., 2024)) on a large margin (+12.49 and +19.03 points F-score improvement on Single-Source and Multi-Source subsets, respectively) since the accurate object segmentation masks are obtained at the segmentation stage and the association module correctly assigns the high similarity to the sounding object mask. Notably, our approach also outperforms the supervised baseline, AVS (Zhou et al., 2022) (65.47 vs. 57.80) on the Multi-source subset with the same ResNet-50 architecture as a visual backbone, while our approach does not require ground-truth audio-visual masks during the training. Moreover, our approach's performance is on par with recent sound source localization approaches on the VGG-SS/Extended VGG-SS approaches, showing robustness against non-visible sounding sources or no-sounding object inputs.

**Audio-Visual Semantic Segmentation.** Table 2 summarizes the quantitative results of the audio-visual semantic segmentation task. Since there is no prior work that does not use ground-truth audio-visual semantic masks during the training, we show the results of the supervised audio-visual semantic segmentation approach (Zhou et al., 2023) as well as video object segmentation models (Mahadevan et al., 2020; Yang et al., 2021) which also require ground-truth semantic masks during the training. These approaches are trained on the AVSBenchSemantics (Zhou et al., 2023) dataset. This experiment additionally uses the CLIP with ViT-B/16 (Dosovitskiy et al., 2021) backbone for our approach. From Table 2, the mIoU of our approach is on par with the supervised baseline (20.18 vs. 20.60 on the ResNet-50 visual backbone) even though our approach does not require an audio-visual semantic mask during the training. Moreover, using a larger backbone (ViT-B/16) in our approach further improves the performance from 20.60 to 25.52.

Table 3: Ablation study of the multi-source-aware (MSA) training on single-source, multi-source, and audio-visual semantic segmentation (AVSS) subsets. MICL loss is employed when the audio mixing is not applied, and MSA-MICL loss is employed when the audios are synthetically mixed during the training.

| Method | Single-Source | | Multi-Source | | AVSS | |
|---|---|---|---|---|---|---|
| | mIoU ($\uparrow$) | F-score ($\uparrow$) | mIoU ($\uparrow$) | F-score ($\uparrow$) | mIoU ($\uparrow$) | F-score ($\uparrow$) |
| w/o MSA | 63.48 | 79.78 | 43.23 | 63.79 | 20.38 | 23.32 |
| w/ MSA | **65.31** | **81.52** | **47.39** | **65.47** | **20.60** | **23.56** |

Table 4: Segmentation accuracies according to the number of the training samples on single-source, multi-source, and audio-visual semantic segmentation (AVSS) settings.

| Training Data Size | Single-Source | | Multi-Source | | AVSS | |
|---|---|---|---|---|---|---|
| | mIoU ($\uparrow$) | F-score ($\uparrow$) | mIoU ($\uparrow$) | F-score ($\uparrow$) | mIoU ($\uparrow$) | F-score ($\uparrow$) |
| 50k | 58.89 | 77.12 | 37.12 | 56.42 | 19.02 | 21.56 |
| 100k | 63.47 | 80.12 | 43.56 | 63.41 | 20.31 | 23.23 |
| 144k | **65.31** | **81.52** | **47.39** | **65.47** | **20.60** | **23.56** |

## 3.5 ABLATION STUDY

**Local-Global Mask Embedding Representation.** The ablation study of the input mask representation is summarized in Fig. 4. In these graphs, we change the mixing weight $\lambda$ in Eq. (1) from 0.0 to 1.0 with 0.2 interval and train the model. The table shows the effectiveness of the proposed local-global representation since using both information achieved the highest segmentation performance on Single-Source and Multi-Source subsets.

**Multi-source-aware Training.** The ablation study of applying synthetic audio mixing augmentation and employing MSA-MICL loss (Eq. (8)) is summarized in Table 3. The MICL loss is employed when the augmentation is not applied. Synthetically generating the multi-source audio waveforms and the loss function that considers the multiple positive samples improves the performance on the multi-source and single-source subsets.

**Number of Input Masks.** Fig. 5 shows the ablation study of the number of object masks ($N$) detected from MViT (Maaz et al., 2022). If $N$ is small, the sounding objects may not be inputted to the audio-mask association module, while more non-sounding objects are inputted to the module if the number of $N$ is large. The table shows that the segmentation accuracy on all subsets is improved along with the increase in the number of object masks.

**Scale of the Training Dataset.** Table 4 summarizes the segmentation accuracy changes when the number of training samples of the audio-mask association module is changed. From Table 4, increasing the number of training samples also improves the segmentation accuracy, showing the importance of a variety of unlabeled data in the training dataset.

## 4 RELATED WORK

**Sound Source Localization.** The sound source localization task aims to predict the location of sounding sources in the images. The sound source localization works can be categorized into weakly-supervised (Qian et al., 2020; Senocak et al., 2022; Mo & Raj, 2023) and unsupervised approaches (Senocak et al., 2018; Tian et al., 2018; Oya et al., 2020; Chen et al., 2021; Fedorishin et al., 2023; Lin et al., 2023; Park et al., 2023; 2024). The weakly-supervised approaches utilize the category labels of the sound in addition to the audio-image pairs during the training. On the other hand, the unsupervised approaches are solely trained on audio and image pairs to associate between different modalities. EZ-VSL (Mo & Morgado, 2022a) employs a multiple-instance contrastive learning framework to align the embeddings between audio and a set of grid-level visual feature embeddings. Recently, ACL-SSL (Park et al., 2024) utilizes pretrained CLIPSeg (Lüddecke & Ecker, 2022) and replaces the text embedding in CLIPSeg with audio embedding.

However, since most approaches align the embeddings between audio and grid-level visual features extracted from the input images using vision encoders, they can only roughly localize the sounding object and fail to segment the sounding object at a pixel level. In contrast to the prior works, we align the embeddings between audio and the *pre-generated object masks* to achieve pixel-level audio-visual segmentation without ground-truth audio-visual mask annotations. The most relevant work is ProSelectNet (PSN) (Xuan et al., 2022). Although PSN and ours employ object proposals, one major difference is that PSN selects proposals via the global audio response map (GRM), whereas our approach directly associates proposals with audio. The two-staged association (response map → select) has drawbacks: (1) errors in GRM propagate to the second stage, and (2) it heavily relies on the coarse GRM, resulting in failures to localize the small-sized objects. Our single-stage association approach, which directly associates audio and masks regardless of size, overcomes these drawbacks.

**Audio-Visual Segmentation.** The audio-visual segmentation task, which requires the model to predict whether each pixel corresponds to the given audio, and the benchmarks (AVSBench and AVSBenchSemantics) (Zhou et al., 2022; 2023) are newly proposed. They provide videos along with the audio and the ground-truth audio-visual masks, and the model is trained with the existence of ground-truth masks. The succeeding works (Zhou et al., 2022; Gao et al., 2023; Liu et al., 2023b; Wang et al., 2024a; Mo & Tian, 2023; Chen et al., 2023; Li et al., 2023; Yang et al., 2024; Liu et al., 2024b; Yang et al., 2024; Wang et al., 2024a; Seon et al., 2024; Wang et al., 2024b) focus on the network architecture, such as cross-modal feature extraction (Zhou et al., 2022; Gao et al., 2023; Liu et al., 2023b; Wang et al., 2024a; Mo & Tian, 2023; Chen et al., 2023; Liu et al., 2023a), and object-aware audio-query (Li et al., 2023). The audio-visual segmentation approaches (Mo & Tian, 2023; Liu et al., 2024b; Yang et al., 2024; Wang et al., 2024a; Seon et al., 2024; Wang et al., 2024b) proposed to effectively leverage the recent progress in the pretrained foundation segmentation models, such as SAM, to improve the segmentation accuracy. Most approaches use the audio signal as an input prompt instead of visual geometric prompts, such as points or bounding boxes.

However, the previous audio-visual segmentation approaches, including SAM-based approaches (Mo & Tian, 2023; Liu et al., 2024b; Yang et al., 2024; Wang et al., 2024a; Seon et al., 2024; Wang et al., 2024b) require annotated audio-visual masks during the training, and the annotated training dataset limits the scalability of the model. In contrast to the previous approaches, we leverage the strong performance of pretrained segmentation foundation models by generating audio-agnostic masks, and the model is trained to associate between audio and a set of object masks on unlabeled audio-video dataset (Chen et al., 2020).

## 5 CONCLUSION

In this paper, we propose a framework that decouples audio-visual segmentation and semantic segmentation tasks into multiple distinct stages: (1) object segmentation solely from an input image, agnostic to class and audio, (2) association between input audio and object masks, and (3) mask classification. Throughout this decoupling, we leverage pretrained vision foundation models to achieve audio-visual segmentation tasks without relying on ground-truth audio-visual masks for model training. Specifically, we employ a multiple-instance contrastive learning framework and train the audio-mask association module in an unsupervised manner. We introduce local-global mask embedding and multi-source audio-aware training to further enhance performance. Experimental results verify that our approach achieves state-of-the-art performance on benchmarks without using ground-truth audio-visual masks.

**Limitation.** The input images are used as visual data following conventional sound source localization approaches. However, segmenting only the sounding object becomes challenging when multiple objects with the same category exist in the image. Therefore, investigating the propagation of temporal information of the object masks is considered for future work.

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

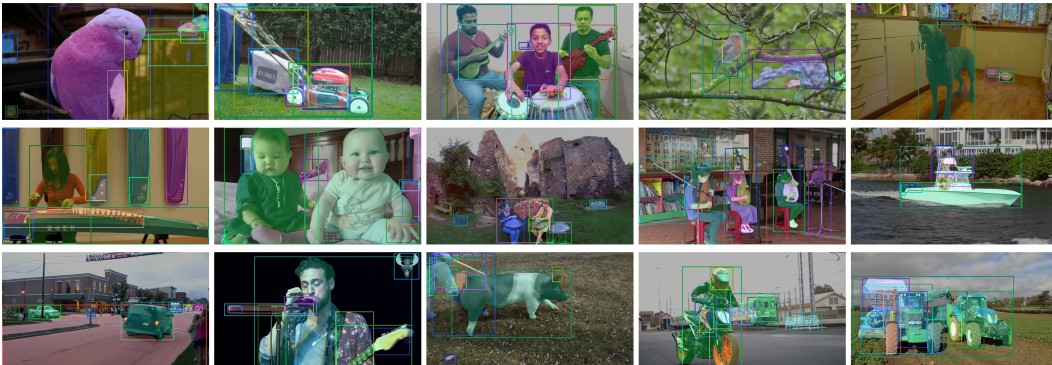

Figure 6: Example of detected object masks on AVSBench (Zhou et al., 2022) at the first segmentation stage. The detected object masks are visualized in different colors, and only 10 masks with high detection confidence values are shown for the visualization.

# A APPENDIX

## A.1 IMPLEMENTATION DETAILS

The mel spectrum is input to VGGish (Hershey et al., 2017) model, which is pretrained on AudioSet (Gemmeke et al., 2017). The dimensions $d$ of the audio and mask embedding vectors are set to 128, and the number of layers of the Transformer in the mask encoder is set to 4. We employ GELU (Hendrycks & Gimpel, 2016) and Layer Normalization (Ba et al., 2016) as an activation function and normalization layer, respectively. We employ AdamW optimizer (Loshchilov & Hutter, 2019) with the initial learning rate $1e^{-4}$ and weight decay 0.01. The learning rate is linearly decayed throughout the training, and the number of training epochs is 30 for all evaluation settings (Single-Source, Multi-Source, and Semantic Segmentation). No data augmentation against visual data is applied. For obtaining text embeddings using CLIP text encoder, we employ templates used on the ImageNet (Deng et al., 2009) experiment used in the original CLIP's zero-shot experiment[4]. The model is trained using a single NVIDIA RTX 3090Ti. The hyperparam were simply found in a standard coarse-to-fine grid search or step-by-step tuning using the validation set in AVSBench (Zhou et al., 2022) benchmark. For the experiments on the Extended VGG-SS dataset, the evaluation requires calculating the confidence score of the predictions. Following the prior work (Mo & Morgado, 2022b), we employ max *cos. sim.* before *min-max norm.* among masks as confidence. Moreover, since VGG-SS and Extended VGG-SS only have ground-truth bounding boxes, we also assign the audio similarity to the detected bounding boxes, not to the object masks.

## A.2 QUALITATIVE RESULTS

## A.3 OBJECT SEGMENTATION

Fig. 6 shows more qualitative visualization of the detected object masks at the segmentation stage. It can be seen that the various objects, including the sounding or sound-irrelevant objects, are detected from the images.

### A.3.1 AUDIO-VISUAL SEGMENTATION

The additional qualitative results of the audio-visual segmentation and semantic segmentation tasks are visualized in Fig. 7.

---

[4]https://github.com/openai/CLIP/blob/main/notebooks/Prompt_Engineering_for_ImageNet.ipynb

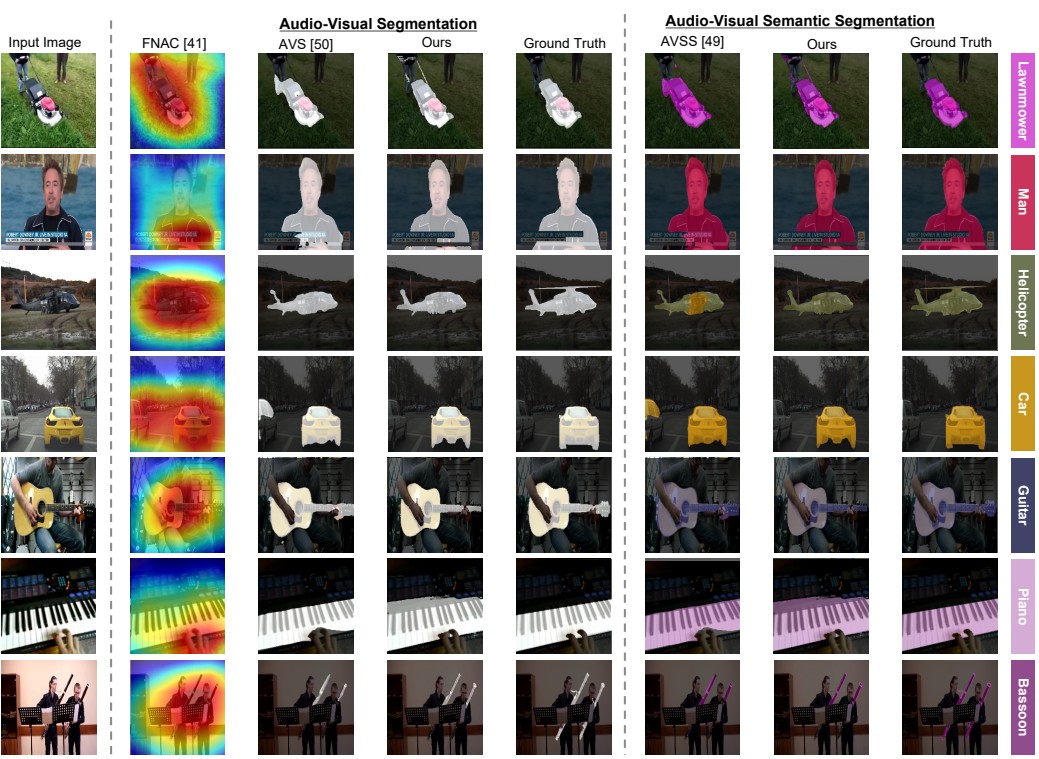

Figure 7: Qualitative results of the audio-visual segmentation and semantic segmentation.

