# OpenReview forum: "Segment, Associate, and Classify: Decoupled Audio-Visual Segmentation Framework"
_ICLR.cc/2025/Conference — ICLR 2025 Conference Withdrawn Submission_

### Official Review · Reviewer_k8Aw · 2024-10-28

**Soundness:** 3
**Presentation:** 2
**Contribution:** 2
**Rating:** 5
**Confidence:** 4

**Summary:**

This paper introduces a new SeAC method for audio-visual (sementic) segmentation task. SeAC operates in three stages: 1) Using only visual frames, it emplys an image segmentation model and SAM to generate pixel-level masks for visual objects. 2) By incorporating the audio, the masks of sounding objects can be identified by evaluating audio-mask feature similarities. 3) The categories of sounding masks are predicted using a paradigm similar to CLIP. The model is trained without pixel-level ground turths, utilizing the proposed multiple sample - multiple instance contrastive learning (MSA-MICL) loss.

**Strengths:**

1. The proposed three-stage method is well-motivated. The studied audio-visual (semantic) segmentation task requires generating pixel-level masks (1) of the sounding objects (2) and predicting their category semantics (3).
2. Most prior works on audio-visual segmentation rely on pixel-level ground truths, whereas the proposed method can be used in an unsupervised manner. The MSA-MICL approach brings obvious improvements.
3. The proposed method demonstrates superior or competitive performances on multiple benchmarks.

**Weaknesses:**

1. I have concerns about the novelty of the proposed method. In particular, the 'Segment' stage uses existing image segmentation models to obtain visual masks. Notably, this has also been employed in prior works, for example, *Prompting Segmentation with Sound Is Generalizable Audio-Visual Source Localizer (AAAI 2024)* ; The 'Classify' stage simply uses existing CLIP model to decide sound source categories; The proposed MSA-MICL is similar to existing EZ-VSL method.
2. One of the main contributions of this paper is the unsupervised contrastive learning. However, it seems that the authors utilize 144k videos from VGGSound for model training. Since the AVSBench datasets are collected using techniques similar to those for VGGSound, there may be a risk of testing data leakage. Moreover, the introduction of MSA-MICL contrastive loss is unclear; this loss will be influenced by the construction of synthetic data, about which details and discussions are not provided.
3. More questions will be given in the next part.

**Questions:**

1. In Eq. (6), the MICL loss contains two items. Could the authors provide an ablation study to explore the impacts of each item?
2. In Eq. (1), the proposed global-local mask embedding integrates the background information as global cues.  However, will the background also include other meaningful visual objects, leading to confusion in embedding?
3. In Line 159, the audio signal is embedded into a unified feature vector. When the audio contains mixed sounds (or the sound changes in temporal segments), how does the method associate the audio with various visual masks by identifying the maximum feature similarity?

---

### Official Review · Reviewer_njqx · 2024-11-01

**Soundness:** 3
**Presentation:** 3
**Contribution:** 3
**Rating:** 5
**Confidence:** 3

**Summary:**

The paper presents a decoupled framework for AVS, which includes three stages:
- Object Segmentation;
- Audio-Mask Association; and
- Mask Classification.

This method stands out by performing segmentation at a pixel level and demonstrating significant improvements over unsupervised and weakly-supervised models while achieving comparable results to supervised baselines.

**Strengths:**

This paper introduces a novel decoupled framework for unsupervised audio-visual segmentation by segmenting, associating, and classifying objects in a sequential process. While each component draws on existing methods, the modular approach allows flexibility in adapting and optimizing individual stages without compromising the overall model structure. The use of multiple-instance contrastive learning (MICL) for associating audio with segmented objects, combined with multi-source audio augmentation, effectively addresses challenges in unsupervised audio-visual learning.

**Weaknesses:**

While the paper achieves impressive performance, it essentially stacks pre-existing modules (e.g., pretrained segmentation and vision-language models) in a decoupled framework, with limited architectural innovation, It feels more like a pipeline.

The figs focus on single-frame segmentation without evaluating time-based alignment in dynamic scenes. This limitation misses a critical aspect of audio-visual synchronization. Continuous multi-frame results or temporal metrics would help verify the framework's ability to manage complex time-dependent audio-visual correlations.

The model implicitly assumes every audio segment is linked to a visible object. In real-world applications, background music or unrelated sounds are common, which may lead to incorrect associations. Implementing a “null correspondence” mechanism or similar approach could help address this limitation.

**Questions:**

- How does the model ensure *temporal alignment* in dynamic scenes?
- Given that many real-world videos contain sounds that do not correspond to visible objects, how does the model address these cases?
- Is there any unsuccessful outcomes? Discussion about those video sample will be helpful.

---

### Official Review · Reviewer_MwWZ · 2024-11-03

**Soundness:** 3
**Presentation:** 4
**Contribution:** 3
**Rating:** 6
**Confidence:** 4

**Summary:**

A model to achieve high-performance unsupervised AVS model.

**Strengths:**

The paper is clearly written and easy to follow.
The performance is comparative.
Unsupervised AVS is an urgent and essential task.

**Weaknesses:**

1. Accumulation error in matching: Could you please provide more examples and statistics related to matching? Since the audio label and visual label may not always align perfectly. In AVS-bench, the segmentation labels can be quite ambiguous, such as "man" and "boy," "car" and "ambulance." Have any analyses been conducted on this issue?
2. Accumulation error in detection: The class-agnostic object detector often detects unwanted objects and assigns incorrect classes. Is there any further analysis on this matter?
3. Further analysis on mask-wise audio similarity: Let's consider the image in Figure 1 as an example. The man in the image can produce not only speech but also sounds like clapping and whistling. How can mask-wise audio similarity help address this issue? In other words, sounds like clapping and whistling can be emitted by multiple different objects, like men and women.
4. Dataset: In the paper, the authors claim that they divide AVSBench into two 5-second clips. Is it fair for other models that take 10s input? Why 5 seconds?
5.  Fair comparison: Can authors provide the comparison of parameters and FLOPS?

**Questions:**

1. Weird masks: Why do the "ours" segmentation samples in Figure 3 appear brighter than others? If all the masks are generated under the same conditions, there shouldn't be such a problem. Therefore, I would expect an explanation; otherwise, I will consider this a minor manipulation of experimental data.
2. Missing essential reference: The important supervised AVS methods should still be considered in the related work.

[1] Chen, Y., Liu, Y., Wang, H., Liu, F., Wang, C., Frazer, H., & Carneiro, G. (2024). Unraveling Instance Associations: A Closer Look for Audio-Visual Segmentation. In Proceedings of the IEEE/CVF Conference on Computer Vision and Pattern Recognition (pp. 26497-26507).

[2] Ma, J., Sun, P., Wang, Y., & Hu, D. (2024). Stepping stones: A progressive training strategy for audio-visual semantic segmentation. arXiv preprint arXiv:2407.11820.

[3] Chen, Y., Wang, C., Liu, Y., Wang, H., & Carneiro, G. (2024). CPM: Class-conditional Prompting Machine for Audio-visual Segmentation. arXiv preprint arXiv:2407.05358.

[4] Guo, R., Qu, L., Niu, D., Qi, Y., Yue, W., Shi, J., ... & Ying, X. (2024). Open-Vocabulary Audio-Visual Semantic Segmentation. arXiv preprint arXiv:2407.21721.

[5] Sun, P., Zhang, H., & Hu, D. (2024). Unveiling and Mitigating Bias in Audio Visual Segmentation. arXiv preprint arXiv:2407.16638.

---

> ### Comment · Reviewer_MwWZ · 2024-11-26
> **Any discussion?**
>
> I am expecting the authors to provide their results and explanations.
>
> Otherwise, I need to reconsider my rating.

---

### Official Review · Reviewer_qix8 · 2024-11-04

**Soundness:** 2
**Presentation:** 2
**Contribution:** 2
**Rating:** 5
**Confidence:** 4

**Summary:**

This paper develops a method for audio-visual segmentation that doesn't require ground-truth audiovisual masks for training. The proposed approach ties together multiple systems, object detection, mask segmentation, audio-visual association and a mask classification, to create the overall audio-visual segmentation system. Along with since-source sounds, the paper also emphasizes the multiple sound sources segmentation situations. The proposed approach relies heavily on pretrained models. Evaluations are done on established datasets like AVSBench, VGG-SS/Extended. Paper shows quantitative as well as qualitative results on the audiovisual segmentation task.

**Strengths:**

– Ground truth mask labeling for audio-visual segmentation can be pretty tedious. Unsupervised approaches to this problem are interesting to explore.

– The paper pays attention to learning for multiple-sound sources and the training method supports that. Several prior have paid less attention to it but multi-source segmentation are more natural problems to solve.

– The idea of having global and local embeddings to capture sounding objects along with the background context is good. Although, there are some questions on the way it is done in the paper.

**Weaknesses:**

– The paper relies primarily on pre-trained models - object detection, segmentation (SAM), CLIP, VGGish. While a workable system for sounding object segmentation can be created using these strong models,  it does appear less interesting and as more of an assembled system with existing models. Moreover, with such an approach the labeling effort has been shifted from essentially labeling sounding objects to the ones used by these models (not such a bad thing, just less interesting).


– The global local embeddings are essentially obtained through a linear combination of embeddings the detected object and rest of the image.  It’s not very intuitive if we are simply combining them through linear combination, then why would a straightforward CLIP embedding of the whole image not capture all of the same information ? Perhaps some experiment where a CLIP embedding of the whole image replacing f_n, with everything else the same, might shed some light.

– The approach seems to work primarily on a closed set, where the mast classification label set is known (and fixed) a-priori ? That might be restrictive given how reliant this approach is to large pre-trained models, where the model should be able to handle open-set.

– The performance seems to increase with the number of Input Masks, all the way to up to 50 masks. Given the experimental settings mostly has very few (1, 2 or so) sounding objects, this is a bit surprising that even though such large number of sounding objects are used, the performance does not plateau or deteriorate. Would be good to discuss this.

– The details of the MSA-MICL loss – why it’s needed, what’s the intuition and how it’s done etc., could be improved. It’s a bit hard to follow.

– For multi-source case in Fig 4, why does lambda > 0.6 leads to such a massive drop in performance, much more compared to single-source.

– Table 4, is not very informative, especially given that the training data scale is too narrow – from 50k to ~150k. More varied scale, say an order or more change in training data could actually be a bit informative. Otherwise, hard to see what we are trying to infer.

– Some more insights might be helpful.  Example, what happens when there are multiple sounds, with one or more of them not in field of view. Is the model able to localize only the visible sounding object? Does it produce more false positives? Or Two objects in the scene which can produce the same sound.

**Questions:**

Please address the weaknesses above.

---

### Note · Authors · 2024-11-26

**Comment:**

Dear Reviewers,

Thank you all for providing constructive and insightful comments.
After deep consideration, we have decided to withdraw the paper.

Best regards,

**Withdrawal Confirmation:**

I have read and agree with the venue's withdrawal policy on behalf of myself and my co-authors.